# Ethnic variation and the relevance of homozygous RNF 213 p.R4810.K variant in the phenotype of Indian Moya moya disease

**Arun K.[1], C. M. Shafeeque[2], Jayanand B. Sudhir[3], Moinak Banerjee[2], Sylaja P. N.[1]***

**1** Department of Neurology, Comprehensive Stroke Care Program, Sree Chitra Tirunal Institute for Medical Sciences and Technology, Trivandrum, Kerala, **2** Human Molecular Genetics Lab, Rajiv Gandhi Centre for Biotechnology, Trivandrum, Kerala, **3** Department of Neurosurgery, Sree Chitra Tirunal Institute for Medical Sciences and Technology, Trivandrum, Kerala

* sylajapn@hotmail.com

**Data Availability Statement:** All relevant data are within the paper.

**Funding:** This study was funded by Wellcome Trust-Department of Biotechnology (DBT)/India

## Abstract

### Background and purpose

Polymorphisms in Ring Finger Protein 213 (RNF 213) gene have been detected to confer genetic susceptibility to Moya moya disease (MMD) in the East Asian population. We investigated the frequency of RNF 213 gene polymorphism and its association with MMD phenotypes in the Indian population.

### Materials and methods

A case-control study for RNF 213 polymorphism involving 65 MMD patients, 75 parents, and 120 controls were performed. A total of 21 SNPs were screened, of which 17 SNPs were monomorphic. Allelic and genotypic frequency of all polymorphic SNPs were assessed and its association with MMD phenotypes was evaluated.

### Results

The median age of symptom onset was 9 (range 2–17) and 37 years (range 20–58) in paediatric and adult patients respectively. A strong association was observed with RNF 213 rs112735431(p.R4810K) and MMD. Out of 65 patients with MMD, five patients carried the homozygous risk AA genotype. None of the healthy controls carried this homozygous mutation. The mutant allele was detected in MMD patients from Tamil Nadu and North eastern states of India ($p$ = <0.0001). All the patients carrying the mutant allele had an early age of onset ($p$ = <0.0001), higher incidence of bilateral disease (p = <0.002), positive family history (p = 0.03), higher Suzuki angiographic stage ($\geq$3) (p<0.0006) and recurrent neurological events (ischemic strokes and TIAs) (p = <0.009).

### Conclusion

The homozygous rs112735431(p.R4810K) variant in RNF 213 variant not only predicts the risk for MMD but can also predict the phenotypic variants.

Alliance (IA) as part of Research Training Fellowship. (https://www.indiaalliance.org/) with grant number IA/R/16/2/502992 dated 28/11/2017 received by Arun K. The funders had no role in study design, data collection and analysis, decision to publish, or preparation of the manuscript.

**Competing interests:** The authors have declared that no competing interests exist.

## Introduction

Moya moya disease (MMD) is a rare chronic cerebrovascular disorder characterized by progressive bilateral occlusion of the supra-clinoid internal carotid artery (ICA) and its main branches with associated development of fine collateral networks, adjacent to the site of occlusion in the deep areas of the brain [1]. There is marked heterogeneity in the clinical manifestations of MMD, ranging from asymptomatic patients to those with recurrent strokes and intracranial haemorrhage [2]. There also exist numerous differences in the clinical presentation between young children and adults pointing to a genetic basis for different clinical phenotypes. Besides, a strong regional difference has also been noted, with a high occurrence of MMD in East Asian countries particularly Japan, China, and Korea [1, 3]. Studies have demonstrated that single nucleotide polymorphisms in the RNF 213 gene had a strong association with the onset of MMD in both familial and sporadic cases [4, 5]. In the East Asian population, the founder variant RNF 213 p.R4810K was much more frequently found in MMD patients (Japanese, 90.1%; Korean, 78.9%; Chinese, 23.1%) than the general population (Japanese, 2.5%; Korean, 2.7%; Chinese, 0.9%) [5, 6]. RNF 213 p.R4810K was found to be absent from control individuals as well as Caucasian MMD cases [5, 7]. The homozygous mutation of the p.R4810K was associated with an earlier age of onset and a more severe disease than the heterozygous variant [8–11]. However, non-p.R4810K mutations have also been identified in the RNF 213 which include D4013N in Caucasian patients and E4950D and A5021V in Chinese patients [5].

The genotypic characteristics of Indian MMD patients have not been studied systematically to date. The study aims to determine the genetic characteristics of Indian MMD patients by sequencing the 17-q.25-ter' region and to clarify whether RNF 213 is potentially associated with clinical phenotypes.

## Materials and methods

### Study population

This study was approved by the ethics committee of Sree Chitra Tirunal Institute for Medical Sciences and Technology, India. The IEC number are as follows IEC/1150 dated 26/12/2017. All participants were included in the study only after receiving their written informed consent. Hard copies of the consents are available with the ethics division of our institute. Patients with MMD attending the outpatient or inpatients of Comprehensive stroke care program of the Department of Neurology, Sree Chitra Tirunal Institute for Medical Sciences and Technology (SCTIMST), Trivandrum from January to December 2018 were recruited in this study. The MMD diagnosis was confirmed by digital subtraction angiography (DSA), based on the criteria of the Research Committee on the Pathology and Treatment of Spontaneous Occlusion of the Circle of Willis; Health Labour Sciences Research Grant for Research on Measures for Intractable Diseases, Japan. A total of 65 MMD patients, 75 parents, 1 son of the affected person, and 120 controls were enrolled in the study. Twenty-five complete patient trios were available for the study. The family samples were included to interpret the pattern of inheritance of the associated variants. Control participants had no typical MMD symptoms in the form of stroke, TIA or seizures and they were not screened by conventional DSA, MR angiography (MRA), CT angiography(CTA), or other tests. The clinical records of all previously diagnosed cases were examined including hospital charts, clinical notes and imaging studies. The medical history, native place, age of onset, symptoms at onset and angiographic staging were reconfirmed. A detailed family history was taken. Recurrent events were defined as events in the form of ischemic events, seizures or haemorrhagic events that occurred after the initial

presentation. The angiographic stage was evaluated according to the Suzuki classification. The study was approved by the Institutional Ethics Committee.

## DNA extraction and single nucleotide polymorphism genotyping

After obtaining the informed consent, 10 ml peripheral vein blood was extracted from patients with MMD and normal control participants, placed in EDTA anticoagulant tubes and stored in a freezer at −80˚C until analysis. DNA was isolated using the salting-out method [12]. The extracted DNA was spectro-photometrically quantified and checked for purity at an absorbance of 260nm and 280nm. Resequencing of the 17-q.25-ter' region spanning the RNF 213 region covering 21 SNPs was carried out. The selection of SNPs was based on functional significance, minor allele frequency, and their tagging status. The details of the SNPs selected, PCR primers and in silico functional prediction are shown in "Table 1". The PCR primers were designed using Primer-BLAST and verified by UCSC in silico PCR and synthesised by Sigma-Genosys. Genotyping was done by Gradient PCR amplification at a temperature of 55–65˚C. This was followed up by Sanger sequencing using ABI PRISM Big Dye Terminator v3.1 sequencing kit, (Applied Bio-systems, Foster City, CA, USA) according to the manufacturer's instructions and was run on ABI PRISM3730 Genetic Analyser (Applied Bio-systems, Foster City, CA, USA). Sequence analysis was done using Applied Bio-systems sequence scanner V.1.1.

## Statistical analysis

Demographic and phenotypic observations were analysed using SPSS 22.0 (SPSS Inc., Chicago, IL) and the continuous variables were presented as means ± standard deviation and categorical variables were presented as proportions. Genotype and allele frequencies were computed using the Graph Pad Prism 5.01, (Graph Pad software Inc. San Diego, CA, USA). A p-value of <0.05 was considered significant in all observations. The test of association for rs112735431 RNF 213 risk AA genotype with various phenotype variables such as clinical symptoms, age at onset, unilateral/bilateral, distribution of vasculopathy, imaging characteristics, and Suzuki classification, was done using Chi Square test. Linkage Disequilibrium (LD) plots for controls and patients were generated using Haploview 4.2 (broad.mit.edu/mpg/haploview/). LD plots in cases and control display how non-random association of alleles at two or more loci may differ in cases and control. Functional prediction of the associated SNPs was assessed in silico using regulomeDB (regulome.stanford.edu), Genotype-Tissue Expression (GTEx) portal (www.gtexportal.org/), HaploReg (www.broadinstitute.org/mammals/haploreg). The functional significance of the SNPs was checked for the RegulomeDB rank to assess the possible regulatory effect. RegulomDB rank is derived from RegulomeDB database that annotates SNPs with known and predicted regulatory elements in the intergenic regions of the *H. sapiens* genome [13]. Lower scores indicate increasing evidence for a variant to be located in a functional region. Category 1 variants have equivalents in other categories with the additional requirement of expression quantitative trait loci (eQTL) information. RegulomeDB score is computed based on the integration of multiple high-throughput datasets. GTex portal was assessed for possible gene expression of the allelic variants. GTEx database allow the users to view and download computed eQTL results and provide a controlled access system for de-identified individual-level genotype, expression, and clinical data. HaploReg was also used, which is a tool for exploring annotations of the noncoding genome at variants on haplotype blocks, such as candidate regulatory SNPs at disease-associated loci. Interpreted data is presented only with reference to associated risk SNP.

**Table 1. RNF213 SNPs screened for the study and details from ENSEMBL and dbSNP.**

| †RNF213 ‡SNPs screened | Primer 5→3 | Annealing Temp | Product Size | Regulome DB rank | RegulomeDB rank prediction | Variant | Ancestral Allele | §MAF |
|---|---|---|---|---|---|---|---|---|
| **rs6565666** | ‡‡(F) TTTGCGTGGGCCAGGAGAAGC | 62 | 254 | No Data | | Intron | ¶G | A = 0.21 |
| ‖‖‖rs200418091 | | | | No Data | | Intron | **C | T <0.01 |
| rs370268515 | | | | No Data | | Intron | G | A <0.01 |
| rs367879018 | §§(R) GCTCACGGCTTCAATGATGC | | | 5 | TF binding or DNase peak | Missense | G | A <0.01 |
| rs149136204 | | | | 5 | TF binding or DNase peak | Missense | ††T | C <0.01 |
| **rs6565681** | (F) CACTGGGCATTAAGACTGC | 53 | 243 | 3a | TF binding + any motif + DNase peak | 3' UTR | G | A = 0.353 |
| rs143047595 | (R) CCAGCTCAACTGTCATAGC | | | 2b | TF binding + any motif + DNase Footprint + DNase peak | 3 prime ‖UTR | C | T = 0.01 |
| rs185195273 | | | | 4 | TF binding + DNase peak | 3 prime UTR | G | A = 0.01 |
| rs369097124 | | | | 3a | TF binding + any motif + DNase peak | 3 prime UTR | G | A = 0.01 |
| **rs4889848** | (F) TGAGAAAGTGCAGCGTGC | 55 | 252 | 1f | ***eQTL + TF binding / DNase peak | Synonymous | T | C = 0.35 |
| rs114047543 | | | | 5 | TF binding or DNase peak | Intron | G | A = 0.05 |
| rs368578211 | | | | 5 | TF binding or DNase peak | Missense | G | C<0.01 |
| rs139431747 | (R) CCCATTGTTTGGCAGCACTG | | | 5 | TF binding or DNase peak | Synonymous | G | A = 0.02 |
| rs145282452 | | | | 5 | TF binding or DNase peak | Missense | G | A = 0.01 |
| rs569181011 | | | | NA | | Synonymous | G | A<0.01 |
| **rs112735431** | (F) TGAGGCTGGTAAAGTTCCTG | 58 | 191 | 5 | TF binding or DNase peak | Missense | G | A = 0.0012 |
| rs138228835 | | | | 5 | TF binding or DNase peak | Missense | G | C = 0.02 |
| rs370932670 | | | | 5 | TF binding or DNase peak | Missense | ‡A | G = 0.02 |
| rs61746605 | (R) CCTATGCAGTGATCCTTTCG | | | 5 | TF binding or DNase peak | Missense | C | G = 0.02 |
| rs75053281 | | | | 5 | TF binding or DNase peak | Intron | G | C = 0.10 |
| rs200776946 | | | | 5 | †††TF binding or DNase peak | Intron | A | G = 0.25 |

†RNF 213-Ring finger protein 213

‡SNP-Single nucleotide polymorphism

§MAF-Mean allele frequency

‖UTR- Untranslated region

¶G-Guanine

**C- Cytosine

††T-Thymine

‡A-Adenine

‡‡(F)-Forward

§§(R)-Reverse

‖‖‖rs- Reference SNP

¶¶NA-Not applicable

***eQTL- Expression quantitative trait loci

†††TF binding- Transcription factor binding.

## Results

The clinical and demographic characteristics of 65 patients with MMD and 120 normal controls are summarised in "Table 2". The median age of the first symptom in the paediatric and adult patients was 9 years (range 2–17) and 37 years (range 20–58) respectively. The female to male ratio was 1.2:1. Ischemic stroke (49.2%) was the most common clinical presentation. One

**Table 2. Demographic characteristics of patients with Moya moya disease and healthy controls.**

| Clinical features | No. of patients | |
|---|---|---|
| **Moya moya Disease** | **65** | |
| [††]Median age of symptom onset(years) ([†]IQR) | 8 (1–52) | |
| [**]Female: Male | 1.2:1 | |
| [**]**Domicile** | | |
| Kerala | 17 (26.2) | |
| Tamil Nadu | 38 (58.5) | |
| Others | 10 (15.4) | |
| [**]Family history | 7(10.8) | |
| [**]Consanguinity | 9 (13.8) | |
| [**]**Age of Onset of symptoms (years)** | | |
| 0–10 | 39 (60) | |
| 11–20 | 9 (13.8) | |
| 21–30 | 4 (6.2) | |
| 31–40 | 6 (9.2) | |
| 41–50 | 5 (7.7) | |
| 51–60 | 2 (3.1) | |
| Young onset (<18 years) | 45 (69.2) | |
| Adult onset (≥18years) | 20 (30.8) | |
| [**]**Clinical presentation** | | |
| Cerebral Infarction | 32(49.2) | |
| [‡]TIA | 13(20) | |
| Haemorrhage | 6 (9.2) | |
| Seizure | 17 (26.2) | |
| Headache | 20 (30.8) | |
| Syncope | 2 (3.1) | |
| [††]Median [§]NIHSS on admission (IQR) | 2(15) | |
| [††]Median [‖]m RS on admission (IQR) | 1(4) | |
| [**]**Risk Factor profile and associations** | | |
| Hypertension | 7(10.8) | |
| Diabetes | 4(6.2) | |
| Hyperlipidaemia | 1(1.5) | |
| Trisomy 21 | 2(3.1) | |
| Neurofibromatosis | 1(1.5) | |
| Hereditary Spherocytosis | 1(1.5) | |
| Hypothyroidism | 8(12.3) | |
| [**]**Recurrent events** | 51 (78.5) | |
| [**]**Posterior circulation involvement in DSA** | 18 (27.7) | |
| [**]**Bilateral disease** | 60 (92.3) | |
| [**]**Suzuki's staging** | Right | Left |
| Stage 1 | 5 (7.7) | 3 (4.6) |
| Stage 2 | 0 (0) | 2 (3.1) |
| Stage 3 | 8 (12.3) | 4 (6.2) |
| Stage 4 | 13 (20) | 18 (27.7) |
| Stage 5 | 35 (53.8) | 33 (50.8) |
| Stage 6 | 2 (3.1) | 3 (4.6) |
| **Controls** | **120** | |
| [**]Female/Male | 1.14 | |

(*Continued*)

**Table 2.** (Continued)

| Clinical features | No. of patients |
|---|---|
| [††]Median age in years (IQR) | 31 (20–55) |
| [¶]DSA | Not done |

[†]IQR-Interquartile range

[‡]TIA- Transient ischemic attack

[§]NIHSS-National Institute of Health Stroke Scale

[‖]m RS-Modified Rankin scale

[¶]DSA-Digital subtraction angiography.

[**]Categorical variables are presented as n (%)

[††]Continuous variables are presented as median (IQR).

patient presented with an isolated headache only. Recurrent events happened in 78.5%, of which 56.9% had ischemic strokes and TIAs, 37.3% had seizures and 3.9% had haemorrhagic events. Bilateral disease was observed in 92.3%. Suzuki's angiographic stage 4 or 5 was observed in 78.5% and 73.8% on the left side and right side respectively. The birthplace distribution showed 58.5% originated and continued to live in Tamil Nadu, 26.2% in Kerala, and 15.3% in Northeast India.

A total of 21 SNPs were screened in the study of which 17 SNPs were monomorphic. All the polymorphic SNPs were assessed for their allelic and genotypic frequency and evaluated for their role in association with the disease and its clinical phenotypes. A strong association was observed with RNF 213 variant rs112735431 (p.R4810K) (OR, 36.3; 95% CI, 2.11–626; $p$ = 0.0001) and MMD "Table 3". None of the other SNPs were found to be associated with MMD. rs6565681 and rs488948 were in complete Linkage Disequilibrium. All the patients

**Table 3. Comparison of allelic and genotypic frequencies of SNPs of RNF213 in Moya moya patients and controls.**

| Gene SNPs | Group | Genotype (n/Frequency) | | | p value | Allele (n/Frequency) | | p value | Odds ratio | [†]95%CI |
|---|---|---|---|---|---|---|---|---|---|---|
| | | GG | AG | AA | | G | A | | | |
| rs112735431 | Cases | 60 | 0 | 5 | 0.004 | 120 | 10 | 0.0001 | 36.3 | 2.1–626.0 |
| | | (0.92) | (0.0) | (0.08) | | (0.92) | (0.08) | | | |
| | Controls | 104 | 0 | 0 | | 208 | 0 | | | |
| | | (1.0) | (0.0) | (0.0) | | (1.0) | (0.0) | | | |
| rs6565681 | Cases | 20 | 35 | 10 | 0.52 | 75 | 55 | 0.57 | 1.1 | 0.7–1.8 |
| | | (0.31) | (0.54) | (0.15) | | (0.58) | (0.42) | | | |
| | Controls | 40 | 47 | 17 | | 127 | 81 | | | |
| | | (0.38) | (0.45) | (0.16) | | (0.61) | (0.39) | | | |
| rs4889848 | Cases | 20 | 35 | 10 | 0.52 | 75 | 55 | 0.57 | 1.1 | 0.7–1.8 |
| | | (0.31) | (0.54) | (0.15) | | (0.58) | (0.42) | | | |
| | Controls | 40 | 47 | 17 | | 127 | 81 | | | |
| | | (0.38) | (0.45) | (0.16) | | (0.61) | (0.39) | | | |
| rs6565666 | Cases | 46 | 18 | 1 | 0.84 | 110 | 20 | 1.0 | 1.0 | 0.6–1.9 |
| | | (0.71) | (0.28) | (0.02) | | (0.85) | (0.15) | | | |
| | Controls | 74 | 27 | 3 | | 175 | 33 | | | |
| | | (0.69) | (0.25) | (0.03) | | (0.84) | (0.16) | | | |

[†]95%CI-95%confidence interval, Genotype frequency in bracket.

**Table 4. Frequency of AA risk genotype RNF 213 variant rs112735431 with clinical phenotype.**

| Patient characteristics | | Genotype (frequency%) | | | p-value |
|---|---|---|---|---|---|
| | | GG | AG | AA | |
| **Patients with MMD** | | 60 (92) | 0 | 5 (8) | 0.004 |
| **Normal controls** | | 104 (100) | 0 | 0 | |
| **Age at onset** | Childhood onset (<18 y) | 38 (88) | 0 | 5 (12) | 0.0004 |
| | Adult onset (≥18 y) | 18 (100) | 0 | 0 | |
| **Seizure** | Yes | 14 (87) | 0 | 2 (13) | [†]NS |
| | No | 42 (93) | 0 | 3 (7) | |
| **Stroke** | Yes | 42 (93) | 0 | 3 (7) | NS |
| | No | 14 (88) | 0 | 2 (13) | |
| **Family history** | Yes | 5 (83) | 0 | 1 (17) | 0.029 |
| | No | 51 (93) | 0 | 4 (7) | |
| **Circulation** | Anterior circulation alone | 40 (91) | 0 | 4 (9) | NS |
| | Anterior and Posterior circulation | 16 (94) | 0 | 1 (6) | |
| **Vasculopathy** | Unilateral | 6 (100) | 0 | 0 | 0.0021 |
| | Bilateral | 50 (91) | 0 | 5 (9) | |
| **Recurrent events** | Yes | 9 (82) | 0 | 2 (18) | 0.009 |
| | No | 47 (94) | 0 | 3 (6) | |
| **Regional Ethnicity** | Kerala | 17 (100) | 0 | 0 | 0.004 |
| | Tamil Nadu | 34 (92) | 0 | 3 (8) | |
| **Pan Indian Ethnicity** | Kerala | 17 (100) | 0 | 0 | 0.0004 |
| | Others | 38 (88) | 0 | 5 (12) | |
| **Suzuki stage right** | 0–2 | 12 (100) | 0 | 0 | 0.0006 |
| | 3–6 | 42 (89) | 0 | 5 (11) | |
| **Suzuki stage left** | 0–2 | 6 (86) | 0 | 1 (14) | NS |
| | 3–6 | 48 (92) | 0 | 4 (8) | |

[†]NS-Not significant.

who had the rs112735431 (p.R4810K) risk mutant A allele were found to be in homozygous condition. The association with rs112735431 was observed to be significant even after the Bonferroni correction for multiple-comparison correction. No homozygous mutation was detected in the healthy control samples. TDT test could not be carried out due to the small sample size. In our study, only one loci rs112735431 was found to be associated with the risk allele, which was absent in the control population. GTEx data for the associated SNP rs112735431 was not available in the GTEx Biobank portal and therefore, allelic expression variation could not be interpreted. Among the five homozygous probands, the parents were in the heterozygous condition with a clear documented disease history in one parent suggesting that the disease could be transmitted in an autosomal dominant pattern. The ethnic distribution of the patients showed clustering of homozygous mutant allele in Tamil Nadu and in the North eastern states of India with none of the mutant allele noted in MMD patients from Kerala ($p = <0.0001$). All the patients carrying the homozygous mutant allele of rs112735431 (p. R4810K) were also found to have an early age of onset (≤18 years) ($p = <0.0001$). These risk genotype individuals have a significantly higher incidence of bilateral disease ($p = <0.002$) and increased Suzuki angiographic stage (≥3) ($p<0.0006$) and recurrent neurological events ($p = <0.009$) (Table 4, Fig 1).

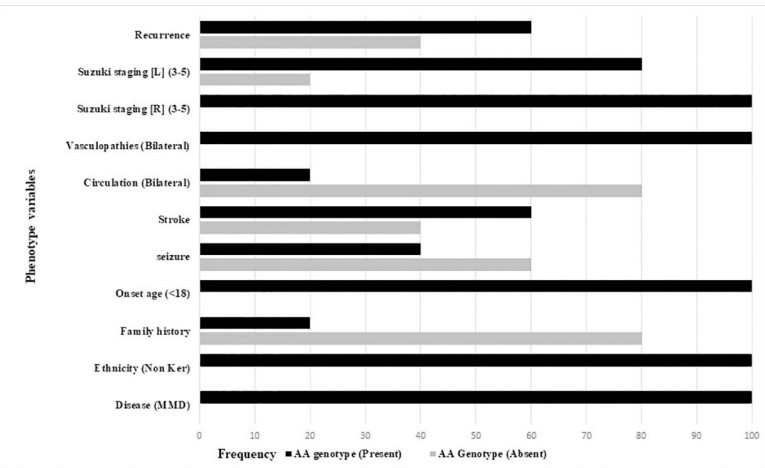

**Fig 1. Frequency of AA risk genotype RNF213 variant rs112735431 (p.R4810K) with clinical phenotypes.**

## Discussion

The study suggests a significant role of RNF 213 rare variant p.R4810K [rs112735431] in MMD and its clinical phenotypes in Indian patients. Population-specific prevalence of risk variants within the Indian subjects, further highlight the need for identifying the ethnic-specific variation, although a larger nationwide study may further highlight the significance of the observations.

The present study revealed a strong association of MMD with RNF 213 variant rs112735431 (p.R4810K) with mutant A allele and AA genotype. No mutant allele was detected in the healthy control population. Studies have shown a higher occurrence of this founder variant in the East Asian population with MMD [5, 6]. Normal populations have also been found to carry this variation at a frequency of 2.5%, 2.7% and 0.9% among Japanese, Korean, and Chinese respectively [5]. Compared to Japanese and Korean patients, the rate of this mutation in Chinese Han MMD patients is lower [13]. In contrast to these observations, we noted this risk variant to be present in only 8% of the Indian patients while it was completely absent in the healthy individuals. Even in Indian patients, it showed ethnic-specific variation. This RNF 213 rare variant significantly increases MMD risk in Korea, Japan and China with an odds ratios (ORs) of 135.63, 338.94 and 14.70 respectively [6]. The incidence of MMD in Europeans is about 1/10 of that found in Japanese [14], and p.R4810K was not identified in Europeans except in one study [15]. Also, among those MMD patients having different descent living in a similar environment, p.R4810K was found in 56% of Asian while none was found in non-Asian descents [7]. Studies have also demonstrated a potential role of non-p.R4810K mutations in the Caucasian population [5] in the MMD pathogenesis. Thus it can be postulated that a multifactorial process might be involved in the etiopathogenesis which includes additional environmental or genetic determinants rather than a definite trait determinant [16]. Even in South East Asian populations, it showed differential penetration where additional variants in RNF 213 were reported in the Han Chinese population [17]. It is interesting to note that even in the present study population we could find a significant ethnic variation with none of the Malayalam speaking population of Kerala carrying the risk variant while all the risk variants are contributed by the individuals from West Bengal and Tamil Nadu. West Bengal populations are known to have gene flow from East Asian genetic lineage. Based on HLA genotyping we have earlier demonstrated that the populations within the south Indian states i.e. Kerala

and Tamil Nadu, have different genetic structure and admixtures among population groups have also been reported between West Bengal and Tamil Nadu [18–20]. Although the rs112735431 did not affect the transcriptional levels or ubiquitination activity of RNF 213, it could reduce the angiogenic activities of induced pluripotent stem cells (iPSC)-derived vascular endothelial cells (iPSECs) in MMD patients [21], and increase the risk of genomic instability in the cells [22].

Our study could demonstrate an autosomal dominant pattern of inheritance as evidenced by the presence of a mutant A allele in heterozygous condition in an affected parent while the proband was homozygous. Though a polygenic inheritance combined with environmental factors has been reported in most cases, various Mendelian patterns of transmission have also been suggested in familial MMD and the disease transmission was noted in multiple generations [23–26]. Mineharu et al. reported 15 large multigenerational pedigrees consistent with an autosomal dominant pattern of transmission with an incomplete penetrance [23].

The presence of mutant A allele was associated with early age of symptom onset of MMD in our patients. Our data correlate well with the reports from the Chinese population [8–11]. Similarly, in a case study conducted among MMD twins with homozygous and heterozygous p.R4810K, the age of disease onset in the homozygote sibling was earlier than that of the heterozygote sibling [11]. Thus it can be suggested that the dosage of p.R4810K alleles was strongly associated with clinical phenotype, even in family members sharing a similar genetic background. However, there are reports of homozygous p.R4810K in unaffected control population [6, 27], and identical twins, with the same dosage of p.R4810K alleles but discordant phenotypes [5]. Therefore, it appears that heterogeneity of the MMD phenotype cannot be explained solely by gene dosage effects. Indeed, environmental factors may also play a critical role in phenotype variation.

The presence of mutant A allele was significantly associated with higher incidences of bilateral disease (p = <0.002), positive family history(p = 0.03), higher Suzuki angiographic stage (p<0.0006) and recurrent neurological events (ischemic events and seizures) (p = <0.009). Miyatake et al. reported that the presence of bilateral vasculopathy was significantly associated with the mutant allele [8]. In our study, 9% with bilateral disease showed the mutant A allele and no mutant allele was detected with unilateral disease. Whether unilateral MMD has a common genetic background or it is a separate entity with similar angiographic characteristics as definite MMD needs to be determined. It has been demonstrated that the presence of a positive family history increased the risk by ten times among Japanese and fifty times among Chinese population [28]. Thus, rs112735431 screening could be utilized as a method to identify asymptomatic patients, especially those who have a positive family history of MMD among Asian descent. Sun et al. [18], conducted a meta-analysis and suggested that the rs112735431 variant should be pursued as a diagnostic screening test particularly in the Asian population, especially when the MMD index case is positive in the family. Our study also demonstrated a higher Suzuki stage in those carrying the mutant allele, suggesting more progressive and severe disease in those carrying the allele. Although the exact molecular mechanisms by which RNF 213 regulates angiogenesis and arteriogenesis remains largely unknown, previous studies have suggested several possible processes that involve two different signalling pathways [29, 30]. The first is a process mediated by the hypoxia-inducible factor-1 (HIF-1) [31] and the second possible signalling pathway may function through caveolin-1 [32]. Serum caveolin-1 levels were found to be decreased in MMD patients and were further decreased in those carrying the RNF 213 p.R4810K variant [33]. Similarly, this mutant allele was associated with recurrent events in the form of ischemic strokes and seizures suggesting a more severe disease. Miyatake et al. [8] observed that the homozygous RNF 213 p.R4810K predicted an early-onset and severe form of MMD. Similarly, a recent study also pointed out a more serious form of the disease

and unfavourable clinical outcome in association with the mutant allele [10]. All these findings prompt us to consider the role played by genetics in the phenotypic characteristics of MMD and to incorporate the genotyping in the routine evaluation of MMD patients so that early diagnosis and early intervention can be planned.

This study has some limitations. This was a single centre study, so there is a chance for selection bias. We investigated the frequency of RNF 213 polymorphism only, however, since MMD is a complex disease with marked genetic heterogeneity and a single gene focus may not sufficiently elucidate the MMD susceptibility. The strengths of our study include; it is the first study from the Indian subcontinent to report the genetics of MMD. We were able to demonstrate the presence of RNF 213 p.R4810K in the Indian population similar to that of the East Asian population. Though the sample size was small, since the frequency of MMD in the Indian population is less as compared to Japan, China and Korea, this population could be considered a representative of the genetics of Indian MMD.

## Conclusions

The homozygous rs112735431(p.R4810K) variant in RNF 213 variant not only predicts the risk for MMD but can also predict the phenotypic variants.

## Author Contributions

**Conceptualization:** Arun K., Moinak Banerjee, Sylaja P. N.

**Data curation:** Arun K.

**Formal analysis:** Arun K., C. M. Shafeeque, Sylaja P. N.

**Funding acquisition:** Arun K., Sylaja P. N.

**Investigation:** Arun K., Jayanand B. Sudhir.

**Methodology:** Arun K., Jayanand B. Sudhir, Moinak Banerjee.

**Project administration:** Sylaja P. N.

**Resources:** Jayanand B. Sudhir, Moinak Banerjee, Sylaja P. N.

**Software:** C. M. Shafeeque, Moinak Banerjee.

**Supervision:** Moinak Banerjee, Sylaja P. N.

**Validation:** Moinak Banerjee, Sylaja P. N.

**Visualization:** Moinak Banerjee.

**Writing – original draft:** Arun K., Sylaja P. N.

**Writing – review & editing:** C. M. Shafeeque, Jayanand B. Sudhir, Moinak Banerjee, Sylaja P. N.

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
