## [Decision Letter · Decision Letter 0]

6 Oct 2020

PONE-D-20-24520

Ethnic variation and the relevance of homozygous RNF213 p.R4810.K variant in the phenotype of Indian Moyamoya disease

PLOS ONE

Dear Dr. Sylaja P.N,

Thank you for submitting your manuscript to PLOS ONE. After careful consideration, we feel that it has merit but does not fully meet PLOS ONE’s publication criteria as it currently stands. Therefore, we invite you to submit a revised version of the manuscript that addresses the points raised during the review process.

The manuscript needs additional editing. Especially, table and figures needs proper lables. The "recurrent events" needs to be defined as well as rational for inclusion of patient groups.

The editor comments should be carfully addressed and corrrected accordingly.

We look forward to receiving your revised manuscript.

Kind regards,

Klaus Brusgaard

Academic Editor

PLOS ONE

Journal Requirements:

Reviewers' comments:

Reviewer's Responses to Questions

**Comments to the Author**

1. Is the manuscript technically sound, and do the data support the conclusions?

Reviewer #1: Yes

Reviewer #2: Partly

2. Has the statistical analysis been performed appropriately and rigorously? 

Reviewer #1: No

Reviewer #2: No

3. Have the authors made all data underlying the findings in their manuscript fully available?

Reviewer #1: Yes

Reviewer #2: Yes

4. Is the manuscript presented in an intelligible fashion and written in standard English?

Reviewer #1: Yes

Reviewer #2: No

5. Review Comments to the Author

Reviewer #1: Very interesting and relevant study about polymorphism in Ring Finger protein RNF213 and MMD. The tables should be more clear the number of SNPs and controls. The confidence interval 3.11-626 is not consistent with the CI in table 3.

Reviewer #2: Moyamoya disease is rare and this group has provided some important findings in an Indian population. There are several issues with this work however that need to be address prior to publication.

-the authors need to copy edit one final time (spaces, consistency, grammar); lines 207-210 do not make any sense

-need to define 'recurrent events' as they offer data on seizure, stroke and recurrent events - quite confusing

-the authors included syndromic MMD (downs, NF2) and they need to address the rational for including these patients

-if the authors choose to discuss the trio data, there needs to be a table with this data in the manuscript. Does the 'one son of a patient' fit into this data?? Unclear

-Materials and Methods: there needs to be much more included in this section. Simply listing the software packages is not enough - how was the software used? Ref to the salting out method? TDT test? Replicates for PCR and sequencing? Odds ratio, bonferroni? What is the meaning of the pval if bonferroni was performed? Instruments used and suppliers? Who made the pcr primers?

-Table 1: terrible format - should fit the page. What does the bold SNP indicate? What does the Regulome score indicate? What is TM?

-Table 3: what are the two lines of data for? Cases and controls? needs labelling

-Table 4: there seems to be only 104 controls. Kerala is listed twice, numbers don't add up. Again, provide definition of recurrent events vs stroke, seizure.

-figure 1: x axis needs a label, y axis is poor quality, ethnicity (non-Ker) needs to be described

6. PLOS authors have the option to publish the peer review history of their article (what does this mean?). If published, this will include your full peer review and any attached files.

Reviewer #1: No

Reviewer #2: No

---

## [Author Response · Author response to Decision Letter 0]

20 Oct 2020

Response to the academic editor and the reviewer

Dear Editor, 

We would like to thank the reviewers for critically reviewing and giving valuable suggestions. 

Response to Editorial comments

Comment 1: The study investigates MMD in Indian patients. This genotypic characterization of Indian MMD has not been studied systematically until date according to the article, and it also makes the study relevant.

We appreciate the positive comment .

Comment 2:The controls were not screened.

A statement in this regard is already mentioned in the manuscript. Control participants had no typical MMD symptoms in the form of stroke, TIA or seizures, but they were not screened by conventional DSA, MR angiography (MRA), CT angiography(CTA), or other tests, which we understand is a limitation.

Comment 3: 22 SNPs were selected. In table1 there is a list of the SNP’s but it seems that there is only 21. 

We thank the reviewers for pointing this out. In the revised manuscript we have amended this as follows: 

A total of 21 SNPs were screened in the study of which 17 SNPs were monomorphic. 

Comment 4: The methods of Sanger Sequencing was performed. More information about the companies of the kits/primers which has been used will be relevant to mention in the article.

In the revised manuscript we have resolved these issues. 

PCR primers were designed using Primer-BLAST and verified by UCSC In-Silico PCR and sythesised by Sigma-Genosys. 

This was followed up by Sanger sequencing using ABI PRISM Big Dye Terminator v3.1 sequencing kit, (Applied Biosystems, Foster City, CA, USA) according to the manufacturer’s instructions and was run on ABI PRISM3730 Genetic Analyzer(Applied Biosystems, Foster City, CA, USA).

Comment 4: Table 3. I need as a reader some explanation about the selection of the SNPs from table 1 compared to table 3. 

The selection of SNPs was based on functional significance, minor allele frequency, and their tagging status. A statement in this regard is mentioned in the manuscript.

The bolded SNPs were tagged to unbolded SNPs and were also screened using the same set of primers mentioned in the primer column..

Bolded SNPs were also polymorphic in our population while unbolded SNPs were monomorphic. 

A statement in this regard is already mentioned in the manuscript. “All the polymorphic SNPs were assessed for their allelic and genotypic frequency and evaluated for their role in association with the disease and its clinical phenotypes.”

Comment 5: In table 3 the controls are mentioned but now it is 104 + 1 compared to 120 controls from the abstract - how should that be understood. 

Thankyou for the valuable suggestion .In the revised manuscript we have revised the table and now the frequencies are mentioned in bracket. It is not 104 + 1 instead it is genotype number 104 (Frequency is 1.0 i.e 100 percent). 

We had 120 samples but only 104 could provide meaningful genotypes in consensus for all SNPs. 

Comment 6: How you calculated the odds ratio? The confidence interval 3.11-626 for rs112735431 is not consistent with the CI in table 3.

We thank the reviewers for pointing this out. In the revised manuscript we have amended the confidence interval. It is 2.11-626 for rs112735431. Odds Ratio was calculated using GraphPad Prism.

Comment 7: Table 4 just minor things about the seizure and stroke 42 (93) 3(3?). 

We thank the reviewers for the comment. In the revised manuscript we have amended this.

 In the revised version the seizure and stroke 42 (93) 3(7).

Comment 8: The discussion is very interesting about the incidence of MMD in populations and that it should be noted that cases of homozygous p.R4810K in unaffected control populations also are reported and that environmental factors may have influence in the phenotype variation.

A very fine section about the limitation of this study with the opportunity to continue working within this very interesting field of polymorphisms in Ring Finger protein RNF213 in MMD diseases.

The conclusion from the reviewer. Very interesting and relevant study about polymorphism in Ring Finger protein RNF213 and MMD. The tables should be more clear as already mentioned (SNP, controls) but I looking forward to following this interesting field.

We thank the editor for the encouraging remarks.

Response to Reviewers comments

Reviewer #1: 

Comment 1: Very interesting and relevant study about polymorphism in Ring Finger protein RNF213 and MMD. The tables should be more clear the number of SNPs and controls. The confidence interval 3.11-626 is not consistent with the CI in table 3.

In the revised version we have tried to improve the tables to fit in one page. Thanks for pointing the confidence interval issue. We have amended the confidence interval. It is 2.11-626 for rs112735431. Odds Ratio was calculated using GraphPad Prism.

Reviewer #2: 

Moyamoya disease is rare and this group has provided some important findings in an Indian population. 

Comment 1: -the authors need to copy edit one final time (spaces, consistency, grammar); lines 207-210 do not make any sense

We have made the edits 

Comment 2: -need to define 'recurrent events' as they offer data on seizure, stroke and recurrent events - quite confusing

The data on strokes and seizures mentioned in the manuscript are initial event the patient presented with. Recurrent events are defined as events in the form of ischemic events, seizures or haemorrhagic events which occurred after the initial presentation.

Comment 3: -the authors included syndromic MMD (downs, NF2) and they need to address the rationale for including these patients

We fully agree with the reviewer that NF2 with moyamoya is considered moyamoya syndrome. But since it has a genetic basis we included that patient. Downs syndrome is mostly reported to coexist with moyamoya disease and is usually not considered moyamoya syndrome.

Comment 4: -if the authors choose to discuss the trio data, there needs to be a table with this data in the manuscript. Does the 'one son of a patient' fit into this data?? 

Since the study was conducted using case control association design, therefore only probands and the controls were considered for table 3. Trios or parental samples were used only for evaluating parental genotypes for the risk alleles in probands. A statement in this regard is mentioned. 

For the case control evaluation we have used only the affected son (proband) while affected parent was not used in the study. However, the affected parent sample genotype was evaluated to see the pattern of inheritance. Parental samples wherever could be used were used only to identify the pattern of inheritance.

Comment 5: -Materials and Methods: there needs to be much more included in this section. Simply listing the software packages is not enough - how was the software used? Ref to the salting out method? TDT test? Replicates for PCR and sequencing? Odds ratio, bonferroni? What is the meaning of the pval if bonferroni was performed? Instruments used and suppliers? Who made the pcr primers?

Overlapping comments have been addressed in response to academic editorial comments. Unaddressed issues are further addressed here. 

All the softwares along with their http links are mentioned in the text. In the revised version all those software that were used for analysis were included and interpreted. (see highlighted text)

A reference to salting out method is included. “6 Miller SA, Dykes DD, Polesky HF. A simple salting out procedure for extracting DNA from human nucleated cells. Nucleic Acids Res. 1988;16(3):1215. doi:10.1093/nar/16.3.1215”

We had mentioned in the text that ”As per Bonferroni correction for multiple comparisons for case control associations for polymorphic variants should be (0.05/4=0.01). The observation obtained in the study with respect to risk SNP is much lower that the 0.01.

PCR replicates were run randomly and also whenever the sequence or the PCR products gave ambiguous product. Lack of amplification or non-interpretable genotypes were not included in the study.

In the revised version instruments and suppliers are included wherever applicable. These are further highlighted in the text.

Comment 6: -Table 1: terrible format - should fit the page. What does the bold SNP indicate? What does the Regulome score indicate? What is TM?

In the revised version we have modified the table for better clarity. 

The selection of SNPs was based on functional significance, minor allele frequency, and their tagging status. 

The bolded SNPs were tagged to unbolded SNPs and were also screened using the same set of primers mentioned in the primer column..

Bolded SNPs were also polymorphic in our population while unbolded SNPs were monomorphic. All the polymorphic SNPs were assessed for their allelic and genotypic frequency and evaluated for their role in association with the disease and its clinical phenotypes.” We have added these to the manuscript

In the revised version we have rephrased the RegulomeDB score as RegulomDB rank. A detail on ranking pattern is explicitly mention in the link. https://www.regulomedb.org/regulome-help/

In the revised version of the table we have rephrased TM as Annealing temp

Comment 7: -Table 3: what are the two lines of data for? Cases and controls? needs labelling

In the revised version of the table this has been amended and the second line has been removed and the frequencies are mentioned in bracket

Comment 8: -Table 4: there seems to be only 104 controls. Kerala is listed twice, numbers don't add up. Again, provide definition of recurrent events vs stroke, seizure.

This has been corrected into regional which refers to south India (Kerala and Tamil Nadu) and Pan Indian ethnicity refers to Non-Malayalam speaking (Non-Ker) population from rest of the country 

Comment 9: -figure 1: x axis needs a label, y axis is poor quality, ethnicity (non-Ker) needs to be described

X and Y axis has been defined in the figure and Non-Ker has been defined in manuscript

---

## [Decision Letter · Decision Letter 1]

19 Nov 2020

PONE-D-20-24520R1

Ethnic variation and the relevance of homozygous RNF213 p.R4810.K variant in the phenotype of Indian Moyamoya disease

PLOS ONE

Dear Dr. Sylaja P.N.,

Thank you for submitting your manuscript to PLOS ONE. After careful consideration, we feel that it has merit but does not fully meet PLOS ONE’s publication criteria as it currently stands. Therefore, we invite you to submit a revised version of the manuscript that addresses the points raised during the review process.

Please carefully adhere to the suggestions and recommendations put forward by reviewer 2 and edit the manuscript accordingly.  

We look forward to receiving your revised manuscript.

Kind regards,

Klaus Brusgaard

Academic Editor

PLOS ONE

Reviewers' comments:

Reviewer's Responses to Questions

**Comments to the Author**

1. If the authors have adequately addressed your comments raised in a previous round of review and you feel that this manuscript is now acceptable for publication, you may indicate that here to bypass the “Comments to the Author” section, enter your conflict of interest statement in the “Confidential to Editor” section, and submit your "Accept" recommendation.

Reviewer #2: (No Response)

2. Is the manuscript technically sound, and do the data support the conclusions?

Reviewer #2: Partly

3. Has the statistical analysis been performed appropriately and rigorously? 

Reviewer #2: No

4. Have the authors made all data underlying the findings in their manuscript fully available?

Reviewer #2: Yes

5. Is the manuscript presented in an intelligible fashion and written in standard English?

Reviewer #2: No

6. Review Comments to the Author

Reviewer #2: The main issue is what types of stats these authors used in their manuscript - it is NOT enough to simply list the software used. There are many errors in the manuscript and the authors should pay attention to this. Beginning of list: INTRO, 'till date' to 'to date', '17q.25-ter'' to 17-q.25-ter' consistently; M&M, again details of the stats used NOT just the programs, RegulomeDB, the authors really don't describe this; DISCUSSION, 'Kerala carry' to 'Kerala carrying'. 'sample size less' to 'sample size was small', the paragraph starting with 'This study has some limitations...' needs serious editing.

7. PLOS authors have the option to publish the peer review history of their article (what does this mean?). If published, this will include your full peer review and any attached files.

Reviewer #2: No

---

## [Author Response · Author response to Decision Letter 1]

24 Nov 2020

Dear Editor/ Reviewer, 

Thanks for your valuable comments, 

1. In the revised manuscript we have revised the ‘Statistical Analysis’ section with explicit explanation of all statistical tests and functional assessments of the SNPs used. 

2. We have also included the details on RegulomeDB rank and also presented it in the ‘table 1’ for better clarity. 

3. Grammatical errors have been corrected and highlighted in the manuscript. 

4. Edits have been made in the paragraph explaining the ‘limitations of the study’.

Statistical Analysis

Demographic and phenotypic observations were analysed using SPSS 22.0 (SPSS Inc., Chicago, IL) and the continuous variables were presented as means ± standard deviation and categorical variables were presented as proportions. Genotype and allele frequencies were computed using the Graph Pad Prism 5.01, (Graph Pad software Inc. San Diego, CA, USA). A p-value of <0.05 was considered significant in all observations. The test of association for rs112735431 RNF 213 risk AA genotype with various phenotype variables such as clinical symptoms, age at onset, unilateral/bilateral, distribution of vasculopathy, imaging characteristics, and Suzuki classification, was done using Chi Square test. Linkage Disequilibrium (LD) plots for controls and patients were generated using Haploview 4.2 (broad.mit.edu/mpg/haploview/). LD plots in cases and control display how non-random association of alleles at two or more loci may differ in cases and control. Functional prediction of the associated SNPs was assessed in-silico using regulomeDB (regulome.stanford.edu), Genotype-Tissue Expression (GTEx) portal (www.gtexportal.org/), HaploReg (www.broadinstitute.org/mammals/haploreg). The functional significance of the SNPs was checked for the RegulomeDB rank to assess the possible regulatory effect. RegulomDB rank is derived from RegulomeDB database that annotates SNPs with known and predicted regulatory elements in the intergenic regions of the H. sapiens genome[13]. Lower scores indicate increasing evidence for a variant to be located in a functional region. Category 1 variants have equivalents in other categories with the additional requirement of expression quantitative trait loci (eQTL) information. RegulomeDB score is computed based on the integration of multiple high-throughput datasets. GTex portal was assessed for possible gene expression of the allelic variants. GTEx database allow the users to view and download computed eQTL results and provide a controlled access system for de-identified individual-level genotype, expression, and clinical data. HaploReg was also used, which is a tool for exploring annotations of the noncoding genome at variants on haplotype blocks, such as candidate regulatory SNPs at disease-associated loci. Interpreted data is presented only with reference to associated risk SNP.

---

## [Editor Report · Decision Letter 2]

1 Dec 2020

Ethnic variation and the relevance of homozygous RNF213 p.R4810.K variant in the phenotype of Indian Moyamoya disease

PONE-D-20-24520R2

Dear Dr. Sylaja P.N.,

We’re pleased to inform you that your manuscript has been judged scientifically suitable for publication and will be formally accepted for publication once it meets all outstanding technical requirements.

Kind regards,

Klaus Brusgaard

Academic Editor

PLOS ONE
---

## [Editor Report · Acceptance letter]

14 Dec 2020

PONE-D-20-24520R2 

Ethnic variation and the relevance of homozygous RNF 213 p.R4810.K variant in the phenotype of Indian Moya moya disease 

Dear Dr. P.N.:

I'm pleased to inform you that your manuscript has been deemed suitable for publication in PLOS ONE. Congratulations! Your manuscript is now with our production department. 

Kind regards, 

on behalf of

Dr. Klaus Brusgaard 

Academic Editor

PLOS ONE